# Impact of Limited Degree of Freedom Drag Coefficients on a Floating Offshore Wind Turbine Simulation

Arjun Srinivas [1,2], Bryson Robertson [1,*], Jonah Benjamin Gadasi [1], Barbara Gwynne Simpson [3], Pedro Lomónaco [1] and Jesús María Blanco Ilzarbe [2]

1    School of Civil and Construction Engineering, Oregon State University, Corvallis, OR 97331, USA
2    Faculty of Engineering, Universidad del País Vasco/Euskal Herriko Unibertsitatea, 48013 Bilbao, Spain
3    School of Civil and Environmental Engineering, Stanford University, Stanford, CA 94305, USA
*    Correspondence: bryson.robertson@oregonstate.edu

**Abstract:** The worldwide effort to design and commission floating offshore wind turbines (FOWT) is motivating the need for reliable numerical models that adequately represent their physical behavior under realistic sea states. However, properly representing the hydrodynamic quadratic damping for FOWT remains uncertain, because of its dependency on the choice of drag coefficients (dimensionless or not). It is hypothesized that the *limited* degree of freedom (DoF) drag coefficient formulation that uses only translational drag coefficients causes mischaracterization of the rotational DoF drag, leading to underestimation of FOWT global loads, such as tower base fore-aft shear. To address these hydrodynamic modeling uncertainties, different quadratic drag models implemented in the open-source mid-fidelity simulation tool, OpenFAST, were investigated and compared with the experimental data from the Offshore Code Comparison Collaboration, Continued, with Correlation (OC5) project. The tower base fore-aft shear and up-wave mooring line tension were compared under an irregular wave loading condition to demonstrate the effects of the different damping models. Two types of hydrodynamic quadratic drag formulations were considered: (1) member-based dimensionless drag coefficients applied only at the translational DoF (namely *limited-DoF* drag model) and (2) quadratic drag matrix model (in dimensional form). Based on the results, the former consistently underestimated the 95th percentile peak loads and spectral responses when compared to the OC5 experimental data. In contrast, the drag matrix models reduced errors in estimates of the tower base shear peak load by 7–10% compared to the *limited-DoF* drag model. The underestimation in the tower base fore-aft shear was thus inferred be related to mischaracterization of the rotational pitch drag and the heave motion/drag by the *limited-DoF* model.

**Keywords:** floating offshore wind; semi-submersible; OpenFAST; quadratic drag; OC5

## 1. Introduction

Climate change is driving a need for alternative energy technologies [1]. As a result of the long-term temperature goals from the Paris Agreement, governmental policies worldwide are striving towards decarbonization [2]. To meet decarbonization objectives, many policies will rely on the utilization of offshore wind resources as a zero-carbon renewable energy source. Offshore wind is estimated to reach about 40% of the total wind energy production in 2050, with global installed offshore wind capacity expected to improve from 29 GW in 2019 to 1748 GW in 2050 [3]. Floating offshore wind (FOWT), in particular, is anticipated to generate 15% of all offshore wind energy by 2050, accounting for 264 GW [3].

According to the Global Wind Energy Council (GWEC), offshore wind is a low-cost, reliable technology that delivers significant economic benefits from manufacturing through operation, and can be deployed rapidly at scale [4,5]. Currently, the largest existing floating wind farm is Hywind Tampen located in the Norwegian North Sea, with a spar-type

platform having a total capacity of 88 MW. The latest offshore wind projects have a capacity factor (the ratio of the electrical energy actually produced to the electrical energy that would have been produced at continuous rated power operation for the same period) of 40–50%, matching the values of an efficient thermal power plant (gas fired or coal fired) in some regions [6,7]. Floating offshore wind has gained momentum, because deep-water installations are subject to abundant and consistent wind, with at least four times as much ocean surface space available compared to bottom-fixed wind types [3].

To support their design and assessment, reliable numerical models need to be able to adequately represent the hydrodynamic loading of FOWT under realistic sea states. The International Energy Agency (IEA) Wind Technology Collaboration Program (TCP) was founded in 1977 to focus on the planning, execution, research, and development of large-scale wind system projects [8]. Under the IEC Wind TCP Task 23 and Task 30, the Offshore Code Comparison Collaborative (OC3) and its subsequent extensions aim to improve understanding of the global behavior of floating offshore wind, including the assessment of existing numerical modeling tools needed for the analysis, design, and assessment of wind turbine technologies [9]. The OC3 and subsequent OC4 projects verified numerical modeling tools through code-to-code comparisons of different offshore wind systems using wind and/or wave load cases. This comparative analysis has contributed to enhanced comprehensibility of FOWT dynamics and modeling techniques. Additionally, it allowed for better understanding of the impact of various model approximations, leading to an improved standard for FOWT modeling [10].

The Offshore Code Comparison, Collaboration, Continued with Correlation (OC5) project was an extension of this work, wherein mid-fidelity numerical tools were validated by comparing numerical responses against experimental data [11]. A 1:50 FOWT model used in the OC5 experiments was subjected to wind and wave loads simultaneously and individually at the Maritime Research Institute Netherlands (MARIN) offshore wave basin (the full-scale FOWT setup is shown in Figure 1). For the wave-only loading cases, the numerical tools compared with the OC5 experimental data consistently underestimated global forces for tower base fore-aft shear and up-wave mooring line tension (refer to Figure 1c). In the frequency domain, the tower base fore-aft shear and up-wave mooring line tension responses were significantly underestimated at low frequencies outside the linear wave excitation regime, associated with the pitch (0.03 Hz) and surge (0.01 Hz) natural frequencies of the OC5 FOWT.

Robustly characterizing the hydrodynamic response is critical to the reliable design and assessment of FOWT. Underestimates from the numerical models found in the OC5 project could be attributed to experimental model uncertainties and/or numerical model inaccuracies [12,13]; e.g., experimental model uncertainties originate from uncertainties in system excitation, system properties, scale effects, and accuracy and precision of the measurement equipment [14]. Numerical model inaccuracies generally relate to the model approximations of the real system physics, which are dependent upon certain aspects, such as the type of the examined load case, calculation of system non-linearities, the representation of system viscous drag, and the like. These inaccuracies can be diminished through building well-performing computational grids, apart from other considerations. Numerically representing the hydrodynamic response for FOWT is closely linked to the selection of realistic drag force models [15]—both linear and quadratic models. For example, many mid-fidelity numerical models implement quadratic drag in two ways: (1) application at the member level, entailing strip-wise application of the Morison drag term that depends on the dimensionless drag coefficients [16–19], and (2) application of global level drag as a matrix [14,20–24]. The strip-wise drag force calculation accounts for relative changes in velocity between the body and fluid, while the latter matrix approach is based only on the body motion (calculated by multiplying the drag coefficients, in dimensional form, from the drag matrix with the square of the body velocity).

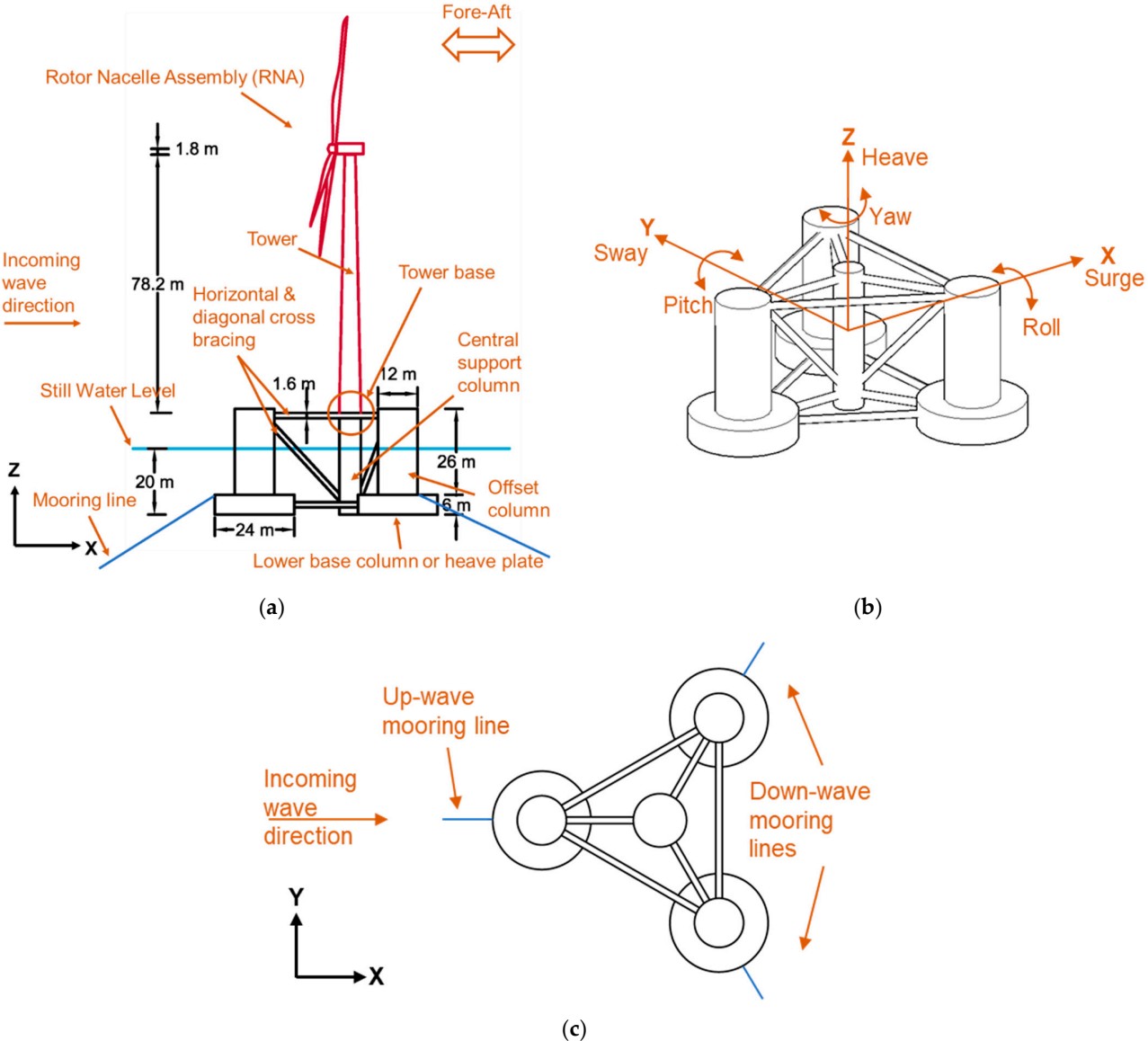

**Figure 1.** OC5 project FOWT model details: (**a**) FOWT full-scale model side view; (**b**) semi-submersible foundation degrees of freedom; (**c**) semisubmersible foundation top view showing the mooring line setup.

The use of a tuned linear and quadratic drag matrix, along with potential flow theory, has shown improved estimates against experimental data, particularly for low-frequency pitch motion and response in past investigations [14,20]. The tuning is performed using the system free decay tests in the desired degree of freedom (surge, heave or pitch). However, a tuned linear (P) and quadratic (Q) drag matrix, or the so-called PQ approach, was not studied in the previous OC5 validation campaign [14]. It is hypothesized that underestimates of the global hydrodynamic loads, particularly the tower base fore-aft shear, is due to mischaracterization of the rotational degree of freedom (DoF) hydrodynamic quadratic drag, because tower base moment-induced shear on a FOWT is strongly influenced by the rotational DoF pitch caused by the wave excitation [13], refer to Figure 1a,b. Similarly, up-wave mooring line tension is strongly influenced by the translational DoF surge, refer to Figure 1b,c. Both platform pitch and surge are excited by differences in hydrodynamic pressure along the wave, which were strong for this FOWT platform [25].

Herein, a numerical model of the OC5 FOWT was studied under wave-only excitation to understand and characterize the impact of hydrodynamic quadratic drag modeling on system response. Two types of hydrodynamic quadratic drag modeling were

employed [11,22]: (1) member-based dimensionless drag coefficient (*limited-DoF*) drag model, and (2) quadratic drag matrix model (in dimensional form). The tower base fore-aft shear and up-wave mooring line tension were analyzed in both time and frequency domains to study the effect of *limited-DoF* drag. To inform the selection of an appropriate hydrodynamic damping model in numerical analyses of FOWT, results were used to: [i] assess the impact of enhanced rotational DoF hydrodynamic quadratic drag on FOWT response, and [ii] characterize how FOWT response can be affected by the source and method of tuning the quadratic drag matrix using the PQ and Faltinsen's methods [26].

## 2. Background on the OC5 FOWT Numerical-Experimental Campaign

For mid-fidelity modeling, the OC5 FOWT can be considered a large volume floating structure, wherein the structure is large relative to the length of the water particle excursion. However, there is a lot of open volume in-between the constituent elements of the platform, and each of them are small in comparison to the wave. It is thus important to consider both the large volume and slender body loading on the semisubmersible platform. This means that there is both a large perturbation of the incident flow field and flow separation at the slender elements of the platform, with the drag forces also important to the FOWT response near resonance. When limiting the effects of hydrodynamic drag forces up to second-order terms, a significant share of linear and quadratic damping originates from radiation damping and viscous drag (from flow separation and skin friction), respectively [27]. Thus, many mid-fidelity models, e.g., OpenFAST, represent the platform hydrodynamics using potential flow-theory-based panel methods augmented with viscous effects using quadratic drag.

### 2.1. Experimental Campaign

Mid-fidelity models currently lack enough accuracy to replace physical model testing [28], necessitating experimental testing and associated numerical validation. Table 1 summarizes the structural and hydrodynamic properties of the OC5 FOWT platform [29]. The OC5 FOWT substructure consisted of a semisubmersible-type platform (refer to Figure 1), which is currently the most common type of FOWT full-scale prototype installation due to its low-cost transportation and installation and wide range of depth feasibility [30–33]. The OC5 semisubmersible platform included three offset columns with a large diameter lower base (acting as heave plates), a central support column below the superstructure (wind turbine and tower), and a series of horizontal and diagonal cross bracing [29]. The platform used a catenary mooring configuration consisting of a chain connecting the floating body to the seabed. For the mooring characteristics, interested readers can refer to the OC5 semisubmersible floating system definition paper [29].

**Table 1.** Structural and hydrodynamic properties of the OC5 semisubmersible platform (full-scale).

| OC5 Phase II Semisubmersible Platform Structural and Hydrodynamic Properties | | |
| --- | --- | --- |
| **Particulars** | **Units** | **Value** |
| Complete system mass | kg | $1.396 \times 10^7$ |
| Draft | m | 20.00 |
| Displacement | $m^3$ | 13,917.00 |
| CM [1] location below SWL [2] | m | 8.07 |
| System roll inertia about CM | kg-$m^2$ | $1.395 \times 10^{10}$ |
| System pitch inertia about CM | kg-$m^2$ | $1.555 \times 10^{10}$ |
| System yaw inertia about CM | kg-$m^2$ | $1.369 \times 10^{10}$ |
| System surge natural frequency | Hz | 0.01 |
| System pitch natural frequency | Hz | 0.03 |

**Table 1.** *Cont.*

| OC5 Phase II Semisubmersible Platform Structural and Hydrodynamic Properties | | |
|---|---|---|
| **Particulars** | **Units** | **Value** |
| Water density, ρ | kg/m$^3$ | 1025.00 |
| Water depth, d | m | 200.00 |
| Displaced water volume | m$^3$ | 13,917.00 |
| Center of buoyancy below SWL | m | $1.32 \times 10$ |
| Static buoyancy force | N | $1.399 \times 10^8$ |
| Hydrostatic restoring in heave | N/m | $3.836 \times 10^6$ |
| Hydrostatic restoring in roll | N-m/rad | $-3.776 \times 10^8$ |
| Hydrostatic restoring in pitch | N-m/rad | $-3.776 \times 10^8$ |

[1] CM is the center of mass of the full system. [2] SWL is the still water line.

### 2.2. Numerical Campaign

OpenFAST is a computationally efficient and well-established floating offshore wind numerical simulation package, that is frequently used for numerical verifications and experimental validation [23,34–38]. Developed by the National Renewable Energy Laboratory (NREL), OpenFAST (formerly FAST or Fatigue, Aerodynamics, Structures, and Turbulence) is one of the most commonly used offshore wind turbine simulation software in academia and industry. Herein, OpenFAST was used to perform the hydrodynamic quadratic drag studies of the OC5 FOWT.

OpenFAST allows for nonlinear coupling between the different dynamics of the FOWT, including the aerodynamics (AeroDyn), hydrodynamics (HydroDyn), control system (ServoDyn), structural dynamics (ElastoDyn), and mooring dynamics (MoorDyn), among others. The coupling is performed through modularization of the software components, wherein each module controls input/output parameters and constraints at the interface between them, representing the different aspects of the FOWT dynamics. Entire modules can be turned on/off depending on the modeling purposes. Herein, the platform hydrodynamics is under study, and thus the wave-only excitation loading using the irregular wave case LC3.3 (refer to Table 2) was used, such that the modules HydroDyn, ElastoDyn, and MoorDyn were only turned on in the numerical simulation.

**Table 2.** FOWT irregular wave-only load case.

| OC5 FOWT Irregular Wave Loading | | | | | |
|---|---|---|---|---|---|
| **Load Case** | **Description** | **Rotor [rpm]** | **Blade Pitch [deg]** | **Wave Condition** | **Simulation Time [min]** |
| LC 3.3 | Operational wave | 0.00 | 90.00 | $H_{m0} = 7.1$ m, $T_p = 12.1$ s, $\gamma = 3.0$ | 176.00 |

Real sea state has waves that are irregular and random in shape, height, speed of propagation, and direction, and such an irregular sea state is usually defined using a wave-frequency spectrum (power spectral density function of the vertical sea surface displacement) [39]. The proposed LC3.3 load case represents the real sea state, defined using the JONSWAP spectrum parameters: significant wave height ($H_{m0}$), peak period ($T_p$), and peak enhancement factor ($\gamma$). Thus, the results presented here can be considered similar to other realistic sea state loads defined by a wave frequency spectrum.

### 3. Hydrodynamic Modelling of FOWT and Drag Forces

Within fluid mechanics, hydrodynamics deals with the study of isovolume fluids, i.e., incompressible and non-dilatable, and forces acting on immersed solid bodies. In general, hydrodynamic forces can be classified by viscous or inviscid effects, both of which contain first- and second-order wave forcing (neglecting higher-order forces). For the OC5 FOWT with catenary mooring, the wave excitation and radiation forces encompass the

inviscid effects, whereas the drag and mooring forces include the viscous effects of the FOWT hydrodynamics.

It should be noted that the semisubmersible platform is considered a rigid body with six DoFs (refer to Figure 1b) and the tower was modeled as elastic in the numerical simulation. The hydrodynamics of the platform were modeled using the potential flow theory-based panel method, along with the addition of quadratic drag through either (1) dimensionless drag coefficients, i.e., Morison drag terms applied using distributed strip theory, or (2) a drag matrix [40]. An external potential flow solver called WAMIT (Wave Analysis at Massachusetts Institute of Technology) was used to model the platform as a 3D diffracting body [41]. Linear potential flow theory solves the radiation-diffraction problem by generating the hydrodynamic coefficients (added mass and radiation damping) in the frequency domain solver WAMIT, which in turn are used to solve the FOWT system dynamics in the time domain solver OpenFAST. WAMIT allows for first- and second-order potential theory solutions from the FOWT submerged body analysis.

The experimental apparatus effects were also considered in the numerical simulation. From earlier validation of the OC5/OC6 numerical models against experimental data, it was found that there is a linear damping effect from the experimental apparatus that needs to be considered [15,27,42]. Additionally, the cable bundle in the experimental setup was found to change the system dynamics from past investigations; hence, a pre-load and linear stiffness in the surge and sway directions needs to be added to the hydrodynamic model [42,43].

Applying Newton's second law of motion to the FOWT subjected to wave-only loading, the governing equations of motion can be formulated according to Equation (1) [17,44]:

$$[M]\ddot{\boldsymbol{\Phi}} = \boldsymbol{F}_{hydostatic} + \boldsymbol{F}_{excitation} + \boldsymbol{F}_{radiation} + \boldsymbol{F}_{mooring} + \boldsymbol{F}^{(2)} + \boldsymbol{F}_{drag} \tag{1}$$

where $\ddot{\boldsymbol{\Phi}}$ is the second time derivative of the generalized floating body motion in each DoF (three translations and three rotations); $M$ is the mass matrix of the FOWT system; $\boldsymbol{F}_{hydrostatic}$ are the hydrostatic forces; $\boldsymbol{F}_{excitation}$ are the first-order wave excitation forces; $\boldsymbol{F}_{radiation}$ are the radiation forces; $\boldsymbol{F}_{mooring}$ are the mooring forces; $\boldsymbol{F}^{(2)}$ are the second-order wave excitation forces; and $\boldsymbol{F}_{drag}$ are the quadratic drag forces.

Note that the wave excitation forces ($\boldsymbol{F}_{excitation}$) consist of a Froude–Krylov force ($\boldsymbol{F}_{FK}$) created by the undisturbed incident wave field, and a diffraction force ($\boldsymbol{F}_D$) created by the interaction of the body with the incident wave field. The pressure fields associated with the undisturbed incident wave and diffracted wave are considered as $\boldsymbol{p}_{FK}$ and $\boldsymbol{p}_D$, respectively. Then, the Froude–Krylov and diffraction forces are obtained by integrating the pressure field over the submerged portion of the body as seen in the equations below.

$$\boldsymbol{F}_{FK} = \iint_{S_B} \boldsymbol{p}_{FK} \boldsymbol{n} dS \tag{2}$$

$$\boldsymbol{F}_D = \iint_{S_B} \boldsymbol{p}_S \boldsymbol{n} dS \tag{3}$$

where $S_B$ is the body surface with $\boldsymbol{n}$ as its normal vector.

The mooring forces ($F_{mooring}$) were calculated in the MoorDyn module within OpenFAST. MoorDyn uses a dynamic mooring line model that accounts for internal axial stiffness and damping forces, weight and buoyancy forces, hydrodynamic forces from Morison's equation, and vertical spring-damper forces from contact with the seabed. The mooring line is discretized into 'L' evenly sized line segments connecting L + 1 node points. Line segments are assumed to be rigid and massless with identical properties of unstretched length, diameter, density, and Young's modulus, while the mass is concentrated at the node points (lumped mass). Hydrodynamic loads are calculated at the node points [45].

Of particular relevance are the second-order wave excitation force and drag force. The OpenFAST hydrodynamic model characteristics used in this paper are detailed in Appendix A.

$F^{(2)}$ represents the second-order wave excitation forces acting on the platform. These forces are proportional to the square of the wave amplitude and are a result of the combination of neighboring frequencies in random waves [46]. It consists of a frequency-dependent, but time-independent mean drift force, difference frequency wave drift force, and sum frequency wave force [16], also known as triads. The last two forces result from the interaction between pairs of wave harmonics, and the forces harmonically change in time with difference or sum frequency, commonly referred to as the wave sub- and super-harmonics [16,47,48]. Drift force can be visualized at the particle level, as the fluid orbital motions do not follow a closed trajectory (first order effect), meaning there is mass transport. The mean and slowly varying (difference frequency loads) drift forces are very relevant for large moored floating structures like the OC5 FOWT. This is because these structures have long natural periods (low natural frequency) in its surge ($\approx$0.01 Hz) and pitch ($\approx$0.03 Hz) motions and the low-frequency second-order forces excite them (particularly the difference frequency component) [46]. It should be noted that the sum-frequency wave forces are generally expected to be more important for tension leg platforms (TLPs) than a catenary moored FOWT [39]. The second-order wave excitation forces are computed in the panel method solver—WAMIT—in the form of quadratic transfer functions (QTFs). These QTFs are imported to HydroDyn solver of OpenFAST and are used to apply the mean and slowly varying drift forces. Equation 4 represents the general form of modeling the second-order wave excitation forces from the QTFs. Herein, full summation and difference QTF are considered in the FOWT numerical simulation.

$$F^{(2)}(t) = \Re\left\{ \sum_{j}^{N} \sum_{k}^{N} a_j a_k Q^{(2)}(\omega_j, \omega_k) e^{i(\omega_j \pm \omega_k)t} \right\} \tag{4}$$

where $a_j$ and $a_k$ are the amplitudes of the wave component $j$ and $k$, respectively, of an irregular sea state with $N$ wave components, $e^{i(\omega_j \pm \omega_k)t}$ is the complex number representation of a wave, $\Re\{*\}$ is the real part of the complex number $*$, $\omega_j$ and $\omega_k$ are the wave frequencies for wave component $j$ and $k$, respectively, $Q^{(2)}(\omega_j, \omega_k)$ is the QTFs, which are six DoF vectors of complex coefficients that are a function of the frequency and the relative heading directions of each wave pair [49].

### 3.1. Drag Force Modeling Methodologies and Representations

The drag force on the floating body, $F_{drag}$, results from the flow field. The steady/unsteady motion of a floating body with sharp corners results in flow separation, creating vortices. There is a transfer of energy from the body motion to the generated vortices. The vortices could be shed even for floating bodies with rounded corners, provided the body is small compared to water particle excursion. The associated force has a damping effect on the body.

The drag force can be applied to the floating body either using Morison's drag term (as shown in Equation (5)) at the member level or by using the global quadratic drag matrix, as mentioned previously.

$$F_{drag} = \frac{1}{2}\rho_w C_D A \left| v - \dot{\Phi} \right| \cdot \left( v - \dot{\Phi} \right) \tag{5}$$

where $\rho_w$ is the density of water, $C_D$ is the member-based dimensionless drag coefficient, $A$ is the projected area (typically the projection of the body on a plane perpendicular to the direction of motion), $v$ is the fluid velocity, and $\dot{\Phi}$ is the body velocity vector.

Generally, the dimensionless drag coefficients (in the Morisons drag term—detailed below) and the quadratic drag matrix are tuned using experimental data. Herein, the drag matrix is tuned using free decay data from both the experimental and OpenFAST numerical models. Two methods of drag coefficient calculation are investigated, namely the Faltinsen's and the PQ method [26,29]. Note that the experimental free decay test data was unavailable. However, drag coefficients calculated using the PQ method on the

experimental free decay were available from OC5 published literature [11]. Different drag force models used in this study are discussed in detail in the subsequent sections.

### 3.1.1. Morison's Drag Force

Application of the member-based drag model calculates the force using the dimensionless drag coefficient, $C_D$, which is determined empirically and is a function of the Reynolds number. The calculation uses the Strip theory, where the body is discretized into a number of strips along the depth with the Morison's drag term applied to each strip, as shown in Figure 2. Under the Strip theory, the cylindrical members are discretized into 1 m long line segments (strips or elements) along the length with the ends of segments representing the nodes. The distributed transverse drag forces are computed at each node and integrated along the depth to obtain the total transverse drag force. Linear wave theory is used to derive the water particle orbit and kinematics at the various depths to aid in the calculation of the drag force [50].

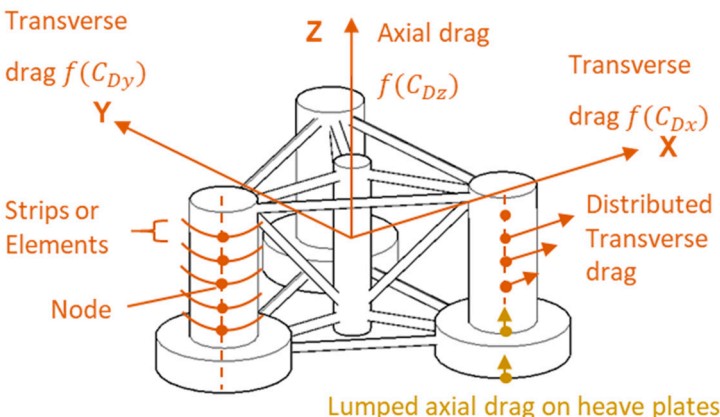

**Figure 2.** Strip-wise discretization of a typical submerged cylindrical body for Morison drag term application.

The dimensionless drag coefficients were applied in the transverse (*x,y*) directions along each strip of a member. The mathematical formulation for the application of Morison drag term along the transverse directions (*x,y*) in the HydroDyn module of OpenFAST is given by Equation (6). Note that HydroDyn does not distinguish the drag between the transverse directions (*x,y*).

$$F_{drag[transverse]} = \frac{1}{2}C_{Dx,y}\rho_w D\left|v_{rx,y}\right|v_{rx,y} \tag{6}$$

where $F_{Drag\,[transverse]}$ is the transverse drag force per unit length along the line segment $[\frac{N}{m}]$, $D$ is the cylinder diameter, '$C_{Dx}$' and '$C_{Dy}$' are the member-based dimensionless drag coefficients along the translational directions $x$ and $y$, respectively, $\rho_w$ is the density of water $[\frac{kg}{m^3}]$, $v_r = v - \dot{\Phi}$ is the relative velocity between body and fluid $[\frac{m}{s}]$.

In the axial (*z*) direction, the dimensionless drag coefficients can be applied either to the top and bottom end nodes of the heave plates (double-sided) or to the bottom end node of the heave plate (single-sided). They are calculated as lumped loads at the member end nodes (axial drag is only calculated at user-defined member end nodes in HydroDyn) [40]. The mathematical formulation for the application of Morison drag term along the axial direction (*z*) on one side of the heave plate is given by Equation (7).

$$F_{drag[axial]} = \frac{1}{2}C_{Dz}\rho_w A_{heave}\left|v_{rz}\right|v_{rz} \tag{7}$$

where $A_{heave}$ is the heave plate face cross-sectional area, '$C_{Dz}$' is the member-based dimensionless drag coefficient along the translational direction $z$.

Two commonly used methods of drag force application using Morison are discussed herein, referred to as Model Typical and Model A, and shown in Table 3. An earlier study

on axial drag coefficient tuning (free decay based) for the Model Typical type of drag force application resulted in only a single-sided drag coefficient applied at the bottom of the heave plate [42]. Model Typical uses a single-valued dimensionless quadratic drag coefficient '$C_D$' applied to all submerged FOWT members. The values used in the former OC5 test campaign are shown in Table 3 [42]. However, several studies have shown that this type of dimensionless quadratic drag tuning results in the mismatch of system motion response at irregular sea states (realistic sea states) [15,27,51,52]. Hence, it was not further studied herein, but is described to show potential modeling choices.

**Table 3.** Drag force application in OpenFAST using the Morison drag term.

| Model Name | Description |
| --- | --- |
| Model Typical | Single-valued transverse $C_D$ (dimensionless) applied to all members in the transverse and axial direction. $C_{Dx} = C_{Dy} = 1.2$; $C_{Dz} = 3.9$ (Single-sided: Applied at the bottom of heave plate) [42]. |
| Model A | Different $C_D$ (dimensionless) applied to different sized (diameter '$D$') members in the transverse and single-valued $C_D$ in axial direction. $C_{Dx,y}\ (D = 1.6\ m) = 0.63$, $C_{Dx,y}\ (D = 6.5\ m) = 0.56$, $C_{Dx,y}\ (D = 12\ m) = 0.61$, $C_{Dx,y}\ (D = 24\ m) = 0.68$, and $C_{Dz} = 4.8$ (Double-sided: Applied at the top and bottom of heave plate) |

Model A, referred to as the *limited-DoF* model, uses member-based dimensionless '$C_D$' value application. As shown in Table 3, Model A only allows entries for the limited translational DoF drag coefficients (transverse and axial member directions). OpenFAST approximates the drag effects in rotation from the values of translational drag coefficients, which assumes a finely discretized slender rotating body, as in the proprietary software ProteusDS and ORCAFlex [53]. However, the FOWT platform consists of non-slender geometries (such as heave plates), where this approximation is invalid.

Model A transverse drag coefficients ($C_{Dx,y}$) use published values from the OC4 phase II definition paper [21], which are flow-regime-averaged $C_{Dx,y}$ values calculated for each member (including heave plates) by interpolating the experimental data found in [54]. Initially, the wave particle velocity is calculated along the depth of the submerged platform column using the linear wave theory. The Reynolds number range for each column member was then formulated to be $Re = 10^5 - 10^7$ for an arbitrary set of periodic sea states (from mild to extreme) defined in the OC4 phase II definition paper. Interpolating the experimental data found in [54], the $C_{Dx,y}$ values were formulated to be in a range between 0.3 and 1.2, for the defined flow regime. The single drag coefficient value for a particular column (as shown in Table 3) can be obtained in two steps. First, the flow-regime-averaged $C_{Dx,y}$ is defined along the depth of the submerged platform column and then this depth-varying $C_{Dx,y}$ is averaged again.

The axial drag coefficient ($C_{Dz}$) for Model A is applied at the top and bottom end nodes of the heave plates, wherein the heave plates are assumed to be flat with flow normal to its face. The $C_{Dz}$ value shown in Table 3 was also published in the OC4 phase II definition paper. It was found by matching a FAST and OrcaFlex coupled simulation with DeepCwind experimental data. Note that the DeepCwind consortium, OC4, and OC5 projects use the same semisubmersible platform.

### 3.1.2. Global Drag Force Matrix

The second model for the drag force representation uses the dimensional hydrodynamic quadratic drag coefficients applied globally using a $6 \times 6$ drag matrix. This application allows for explicit definition of the quadratic drag in all six degrees of freedom (refer to Equations (8) and (9) and can consider both the coupled and uncoupled effects from

different DoFs. Herein, any coupled effects in the quadratic drag matrix were neglected, i.e., only the diagonal terms of the drag matrix are populated.

The global quadratic drag-matrix-based models are referred to as the Model B variants. The drag force application using these models are based on the following equation:

$$F_{Add} = F_0 - [C]\Phi - [B_{linear}]\dot{\Phi} - [B_{quad}]\left|\dot{\Phi}\right|\dot{\Phi} \qquad (8)$$

where $F_{Add}$ denotes the generic addition of preload, stiffness, and damping (linear and quadratic) to the FOWT platform; $F_0$ is the $6 \times 1$ static load vector for preload; $[C]$ is the $6 \times 6$ linear restoring matrix (linear stiffness matrix); $[B_{linear}]$ is the linear drag matrix; $[B_{quad}]$ is the quadratic drag matrix; and $\Phi$ is the generalized motion vector representing the six DoFs. When the FOWT numerical model does not require any force tuning in the form of additional preload, stiffness, and linear drag matrices, the generic additional force $F_{Add}$ transforms to drag force $F_{drag}$ as shown in Equation (9).

$$F_{Add} = F_{drag} = \begin{bmatrix} B_{quad\ xx} & 0 & 0 & 0 & 0 & 0 \\ 0 & B_{quad\ yy} & 0 & 0 & 0 & 0 \\ 0 & 0 & B_{quad\ zz} & 0 & 0 & 0 \\ 0 & 0 & 0 & B_{quad\ Mx} & 0 & 0 \\ 0 & 0 & 0 & 0 & B_{quad\ My} & 0 \\ 0 & 0 & 0 & 0 & 0 & B_{quad\ Mz} \end{bmatrix} \left|\dot{\Phi}\right|\dot{\Phi} \qquad (9)$$

where $B_{quad\ xx}$, $B_{quad\ yy}$, $B_{quad\ zz}$ are the quadratic hydrodynamic drag along the translational DoFs and $B_{quad\ Mx}$, $B_{quad\ My}$, $B_{quad\ Mz}$ are the quadratic hydrodynamic drag along the rotation DoFs.

The drag matrix assembly involves two steps: [i] numerical- or experimental-based free decay test is performed for a particular DoF to define individual wave crest amplitudes, and [ii] constituent linear and quadratic coefficients are extracted from the free decay data and used with the system properties and infinite frequency added mass (the limiting value of added mass at zero period; at this limit there are no diffraction effects).

The Model B variants, namely B1, B2, and B3, differ in the source and method of tuning the drag matrix as shown in Table 4. Model B1 represents more a conventional tuning of the drag matrix based on the experimental OC5 data using the PQ method. Models B2 and B3 implement a novel tuning of the drag matrix based on the Model A decay results using the PQ and Faltinsen's methods, respectively.

**Table 4.** Overview of the numerical models with different drag force application.

| Model Name | Description |
|---|---|
| Model A | Member level drag force application with dimensionless drag coefficients applied to the different sized members in the transverse and axial direction |
| Model B1 | Global level drag force application using a drag matrix assembled from the OC5 experimental free decay data using the PQ method |
| Model B2 | Global level drag force application using a drag matrix assembled from the OC5 numerical Model A free decay data using the PQ method |
| Model B3 | Global level drag force application using a drag matrix assembled from the OC5 numerical Model A free decay data using the Faltinsen's method |

The drag matrix tuning methodologies (PQ and Faltinsen's) differ in the equation of motion representation, constituent coefficient relation, and drag coefficient (with dimension)

calculation. Interested readers can refer to the detailed description of the PQ method in [29], and the Faltinsen's method in [26].

In physical model testing, free decay tests are commonly used to obtain drag coefficients and natural frequencies of the system. Hydrodynamic systems inherently have non-linearities due to the effect of viscous flows [46]. The single degree of freedom equation of motion for such a system with non-linear damping can be represented by Equation (10):

$$(M + A_\infty)\ddot{\Phi}(t) + (B_{linear})\dot{\Phi}(t) + (B_{quad})\left|\dot{\Phi}(t)\right|\dot{\Phi}(t) + (k)\Phi(t) = 0 \tag{10}$$

where $(M + A_\infty)$ is the total mass (or mass moment of inertia) accounting for the system and the infinite frequency added mass component—[kg], [kg.m$^2$]; $B_{linear}$ is the linear drag coefficient—[Ns/m], [Nms/rad]; $B_{quad}$ is the quadratic drag coefficient—[Ns$^2$/m$^2$], [Nms$^2$/rad$^2$]; $k$ is the hydrostatic stiffness coefficient—[N/m], [Nm/rad]. All the coefficients of Equation (10) are in the generalized single degree of freedom motion $\Phi$.

The consecutive crest amplitudes ($\phi_1$, $\phi_2$, and so on) from the free decay test are extracted using the individual wave definition, where two successive down (or up) crossings define the individual wave. Figure 3a show the individual definition using the zero-down crossing method.

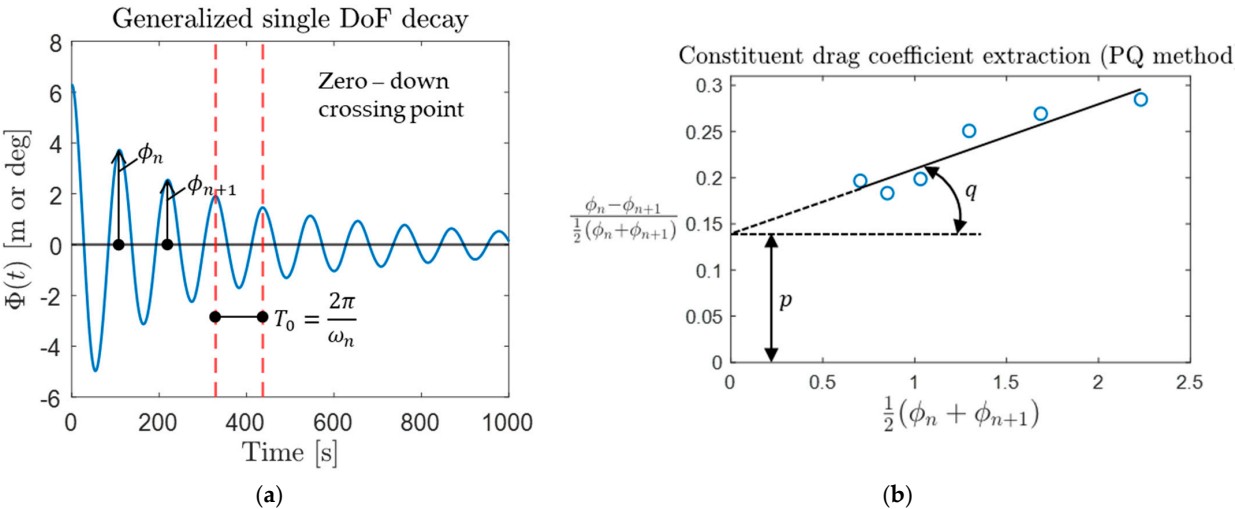

(a)    (b)

**Figure 3.** PQ method constituent coefficient extraction using least square curve fit of the crest amplitudes: (**a**) free decay test showing the wave crest amplitudes; (**b**) straight line curve by least square method to obtain the constituent coefficients 'p' and 'q'.

The constituent linear and quadratic drag coefficients with dimension were then extracted using the consecutive crest amplitude values ($\phi_1$, $\phi_2$, and so on) from the free decay tests using both the PQ and Faltinsen's methods [26,29].

PQ Method

For the PQ method, the constituent drag coefficients, 'p' and 'q,' correspond to the linear and quadratic parts of the system hydrodynamic drag, respectively. The latter coefficient is used to assemble the quadratic drag matrix. The PQ uses the same equation of motion denoted by Equation (10).

In the PQ method, the linear and quadratic constituent coefficients are determined by the relation shown in Equation (11).

$$\frac{\phi_n - \phi_{n+1}}{\frac{1}{2}(\phi_n + \phi_{n+1})} = p + \frac{1}{2}(\phi_n + \phi_{n+1}) \cdot q \tag{11}$$

As shown in Figure 3, the left side of Equation (11) is plotted against the mean motion amplitude term $\frac{1}{2}(\phi_n + \phi_{n+1})$. A straight-line curve fit by least squares is used and the constituent coefficients 'p' and 'q' are calculated from the y-intercept and slope of the fitted straight line.

Faltinsen's Method

The single degree of freedom equation of motion under the Faltinsen's method is expressed in Equation (12):

$$\ddot{\Phi} + (\widetilde{p})\dot{\Phi} + (\widetilde{q})\left|\dot{\Phi}\right|\dot{\Phi} + \omega_n^2\Phi = 0 \tag{12}$$

The coefficients $\widetilde{p}$ and $\widetilde{q}$ denote the linear and quadratic constituent coefficients, respectively. Again, the latter coefficient is used to form the quadratic drag matrix.

Assuming that the damping is constant with respect to the amplitude of oscillation, the linear and quadratic constituent coefficients can be determined from the relation given in Equation (13).

$$\frac{2}{T_0}\log\left(\frac{\phi_{n-1}}{\phi_{n+1}}\right) = \widetilde{p} + \frac{16}{3}\frac{\phi_n}{T_0}\widetilde{q} \tag{13}$$

where $\frac{T_0}{2}$ is the half period between $\phi_n$ and $\phi_{n+1}$ for any $n^{th}$ oscillation. A straight-line curve fit similar to the PQ method is used. The constituent coefficients $\widetilde{p}$ and $\widetilde{q}$ can then be read out from Figure 4 in the form of the y-intercept and slope of the curve fitted line.

$$\widetilde{B_{linear}} = \widetilde{p} \cdot (M + A_\infty); \; \widetilde{B_{quad}} = \widetilde{q} \cdot (M + A_\infty) \tag{14}$$

**Figure 4.** Faltinsen's constituent coefficient extraction using least-square curve fit of the crest amplitudes.

After obtaining the constituent coefficients from the Faltinsen's methodology, multiplying the total mass (consisting of the system mass and the infinite frequency added mass) by the quadratic constituent coefficient gives the quadratic drag coefficient $B_{quad}$ with units of [Ns$^2$/m$^2$] or [Nms$^2$/rad$^2$]. The process is repeated for each DoF motion. The drag matrix with diagonal terms can then be populated.

## 4. Results

The results from the OpenFAST numerical simulations using the four different hydrodynamic quadratic drag models (Models A, B1-B3) are presented below for the OC5 FOWT; subjected to the LC3.3 irregular wave-only loading (refer to Table 2) defined using the JONSWAP spectrum parameters: significant wave height ($H_{m0}$ = 7.1 m), peak period ($T_p$ = 12.1 s), and peak enhancement factor ($\gamma$ = 3.0). Results were analyzed in both the time and frequency domains and compared in terms of global tower base fore-aft shear and up-wave mooring line tension. The time domain history plots for tower base fore-aft shear and up-wave mooring line tension are shown in Appendix B.

*4.1. Time-Domain Results*

The time-domain signal of the tower base fore-aft shear and up-wave mooring line tension were analyzed in terms of mean, coefficient of variation (CoV), and 95-percentile peak load. Mean loads were obtained by dividing the sum of the loads at each time step by the number of time steps. The CoV was calculated by normalizing the standard deviation using the mean, which can be useful for characterizing the dispersion of data around the mean and comparing the variation between different signals whose mean are significantly different from each other [55]. The 95-percentile peak load was obtained by first finding out the local peaks of the time series data and then removing the top 5% of the local peak values data. The top 5% of the local peaks was arbitrarily chosen to remove the data outliers and to be consistent with the original OC5 test campaign [11].

4.1.1. Mean loads and Coefficient of Variation (CoV)

Tables 5 and 6 tabulate the mean loads of the tower base fore-aft shear and up-wave mooring line tension, respectively, for both the experiment and numerical models.

**Table 5.** Mean and Coefficient of Variation (CoV) values for tower base fore-aft shear.

|  | **Experiment** | **Model A** | **Model B1** | **Model B2** | **Model B3** |
|---|---|---|---|---|---|
| Mean loads [kN] | −4.54 | 2.51 | 3.24 | 3.22 | 3.25 |
| Coefficient of variation | 75.70 | 101.36 | 91.83 | 93.84 | 90.71 |

**Table 6.** Mean and coefficient of variation (CoV) values for up-wave mooring line tension.

|  | **Experiment** | **Model A** | **Model B1** | **Model B2** | **Model B3** |
|---|---|---|---|---|---|
| Mean loads [kN] | 1233.85 | 1255.07 | 1254.41 | 1254.87 | 1254.59 |
| Coefficient of variation | 0.15 | 0.12 | 0.11 | 0.11 | 0.10 |

Mean and CoV of Tower Base Fore-Aft Shear

In the case of tower base fore-aft shear, the mean loads for the numerical models were relatively different from the experiment, regardless of the type of damping model. The difference in mean loads between the numerical models and the experiment may be attributed to the experimental setup and its uncertainties as described before in Section 2 (Numerical-Experimental Campaign). The absolute value of the mean is considered in the calculation of the tower base fore-aft shear CoV.

Within the numerical models (Model A, B1-B3), the drag matrix models (Model B1-B3) tended to have similar mean values for tower base fore-aft shear, with differences of ≈0.01 kN to 0.03 kN between each other. The member-based drag coefficients applied in Model A tended to have a slightly smaller mean compared to the drag matrix models. Models using a drag matrix resulted in similar levels of CoV, with Model B2 having the highest degree of dispersion from its mean between the drag matrix models (≈2 to 3).

Mean and CoV of Up-Wave Mooring Line Tension

In the case of up-wave mooring line tension, the mean loads for all the models were large, on the order of 120 MT, with difference in means between the experiment and numerical models of ≈20 kN to 21 kN. Differences are due to the experimental setup, where the effects of bundled cables in the experiment were represented by adjustments in the numerical model with additional preload and stiffness in the surge and sway directions. The CoV values of the experiment and numerical models were similar, on the order ≈ 0.03 to 0.05 difference.

Between numerical models (Model A, Model B1-B3), the mean loads were similar for the up-wave mooring line tension, with differences of ≈0.66 kN (negligible compared to

mean). All the numerical models had similar levels of data dispersion. The drag matrix models, in particular, resulted in CoV very close to each other.

In aggregating these observations, the application of the drag coefficients (dimensionless or not) in the numerical models affects the mean loads and coefficient of variation, and is expected. Similar mean loads and CoV were observed across the drag matrix models for both tower base fore-aft shear and up-wave mooring line tension, because, regardless of the source and method of drag matrix tuning, the global quadratic drag application to the system remains the same for all the drag matrix models.

### 4.1.2. Percent Error Results

The percent error between the numerical and experimental models in terms of the 95-percentile peak load for the tower base fore-aft shear and up-wave mooring line tension are plotted in Figure 5.

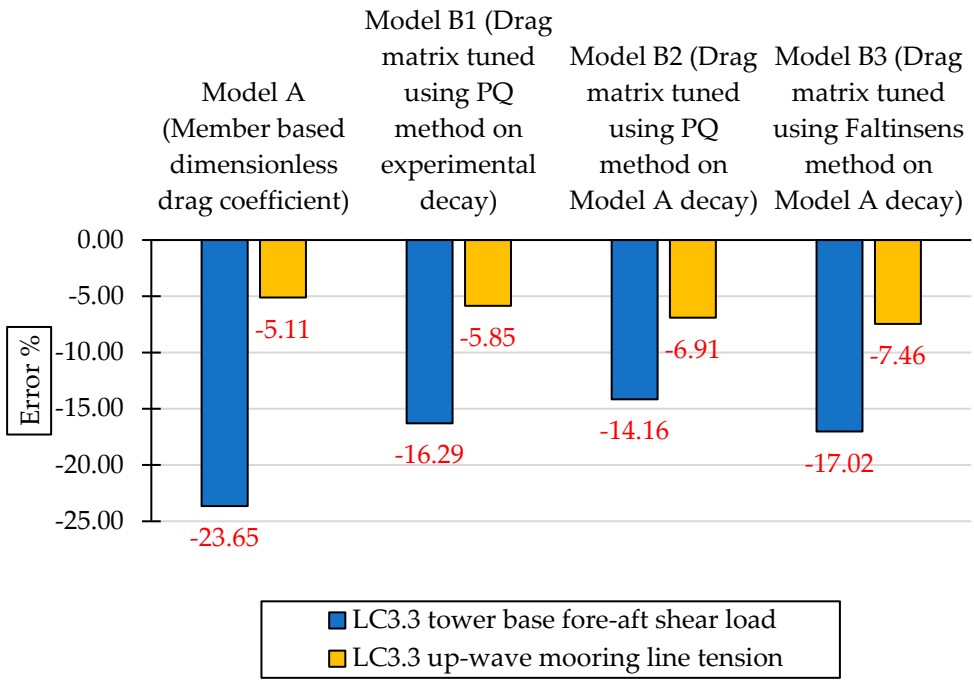

**Figure 5.** LC3.3, 95-percentile peak load error % of the numerical models compared to the experiment.

The negative values on the bar plot indicate underestimation of the loads by the numerical models (Model A, Model B1-B3). Estimates of the tower base shear loads significantly improved when using the drag matrix models (Models B1-B3) compared to the member-based drag coefficient model (Model A)—an improvement (error percent reduction between numerical models) of 7.36%, 9.49%, and 6.63% for Models B1, B2, and B3, respectively, when compared to Model A.

Improved estimates of tower base fore-aft shear in the pitch direction result from improved representation of drag in the rotational DoFs using the drag matrix models. The Model B variants can explicitly define 6 DoF drag to better characterize rotational DoF drag, particularly pitch. In the case of up-wave mooring line tension, the percent error was similar between numerical models, with negligible decrements of 0.74%, 1.8%, and 2.35% for Models B1, B2, and B3, respectively, when compared to Model A. Unlike the tower base for-aft shear, rotational DoF pitch drag does not have a significant impact on the up-wave mooring line tension.

The error percent values brought about by the drag matrix models (B1-B3) for the tower base shear and up-wave mooring line tension are acceptable, given the mid-fidelity level of numerical analysis.

### 4.2. Frequency-Domain-Based Results

In the frequency domain, results are shown using Power Spectral Density (PSD) plots, which are commonly used for random time-domain signals, such as global OC5 FOWT loads, such as tower base shear and up-wave mooring line tension [56]. The PSD, $S_f(\omega)$, is the Fourier transform of the autocorrelation function $R_f(\tau)$ as seen in Equation (16). The PSD plots the signal's power content against frequency, $\omega$ [57]:

$$S_f(\omega) = \int_{-\infty}^{\infty} R_f(\tau) e^{i\omega\tau} d\tau \tag{15}$$

where $R_f(\tau) = \lim_{T\to\infty} \frac{1}{T} f(t) f(t+\tau) dt$; $f(t)$ is the time-domain signal. At $\tau = 0$, $R_f(0) = \lim_{T\to\infty} \frac{1}{T} f^2(t) dt = \frac{1}{2\pi} \int_{-\infty}^{\infty} S_f(\omega) d\omega$. The PSD was calculated by multiplying the signal fast Fourier transform (FFT) by its complex conjugate, resulting in a spectrum where the real part is the amplitude of each harmonic with the signal's units squared, $(*)^2$. The amplitude was then normalized by the frequency step (inverse of total measured duration), $df = \frac{1}{t(N)-t(1)}$; where $N$ is the total signal length. The resulting PSD has units of $(*)^2/Hz$.

#### 4.2.1. Power Spectral Density (PSD) of Platform Motion

The platform motion PSDs in surge, heave, and pitch are plotted in Figure 6a–c, respectively.

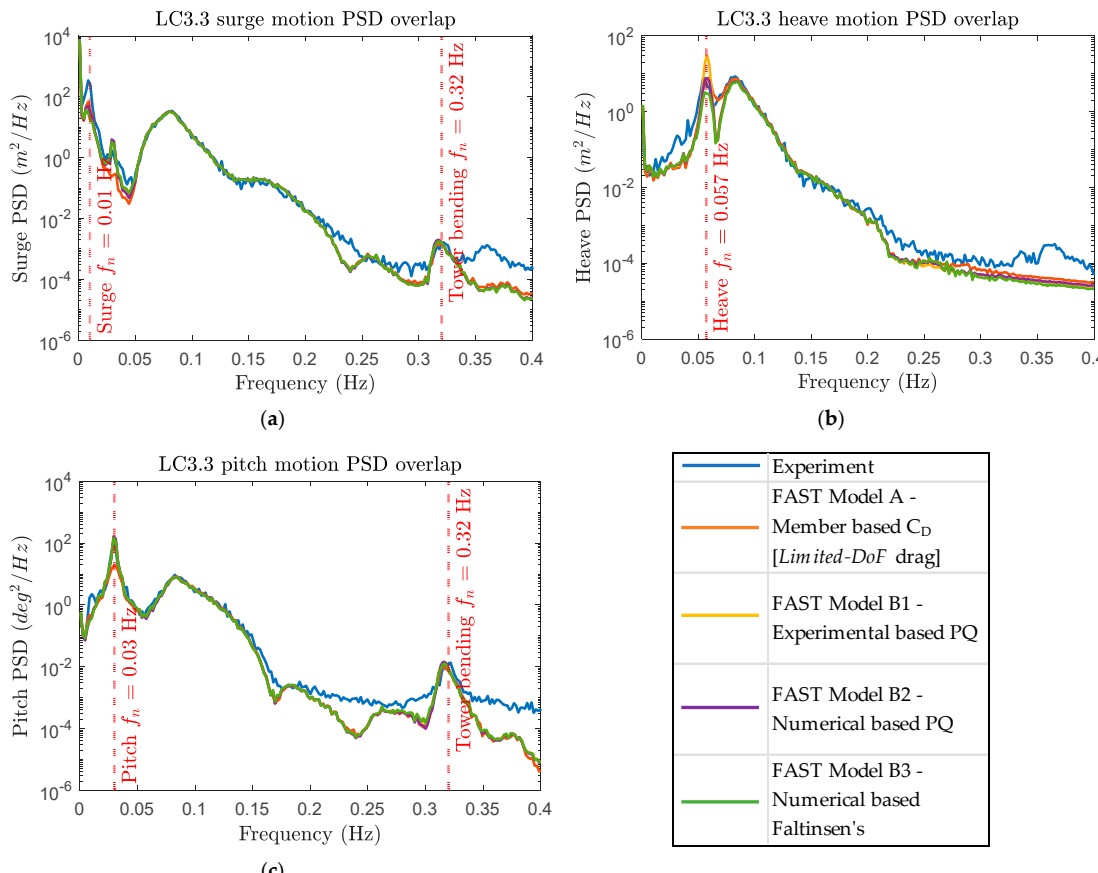

**Figure 6.** OC5 FOWT platform motion power spectral density (PSD) for LC3.3: (**a**) Surge motion PSD; (**b**) heave motion PSD; (**c**) pitch motion PSD.

For the surge motion PSD, the frequency response amplitudes for the system natural frequencies at surge (0.01 Hz), pitch (0.03 Hz), and tower bending (0.32 Hz) are shown for both experimental and numerical model (Models A, B1-B3). All the models underestimated

the response amplitude for surge natural frequency, with the *limited-DoF* Model A better representing the low-frequency surge response compared to the drag matrix models (B1-B3). In contrast, the amplitude associated with surge-induced pitch was not well represented by Model A and was overestimated by the drag matrix models. All the models estimated similar amplitudes for tower bending and were close to the experimental results.

For the heave motion PSD, frequency response amplitudes at the heave natural frequency (0.057 Hz) are discussed across the numerical models. The *limited-DoF* Model A and drag matrix model B2 (drag matrix tuned based on Model A decay using the PQ method) had similar amplitudes for heave. Both models were closer to the experimental results compared to other models because Model A uses drag coefficients tuned to the heave motion based on the experimental data, and Model B2 tunes the quadratic drag matrix based on Model A decay. Models B1 and B3 over- and underestimate the response amplitude at the heave natural frequency, respectively. There is no response amplitude recorded at the tower bending natural frequency by the experimental and numerical models.

For the pitch motion PSD, frequency response amplitudes at the pitch (0.03 Hz) and tower bending (0.32 Hz) natural frequencies are discussed across the numerical models. Model A most underestimated the amplitude associated with pitch, whereas the drag matrix models (B1-B3) consistently remained very close to the experimental values for the pitch response. Thus, the drag matrix models were more capable of characterizing the pitch drag. All the numerical models closely matched the response amplitude at the pitch-induced tower bending natural frequency with the experimental results.

### 4.2.2. PSD of Tower Base Fore-Aft Shear

For the LC3.3 irregular wave loading, Figure 7a shows the whole spectrum plot for tower base fore-aft shear. To show details, sub figures Figure 7b,c zoom into the two main regions of concern, namely near the pitch and tower bending frequencies.

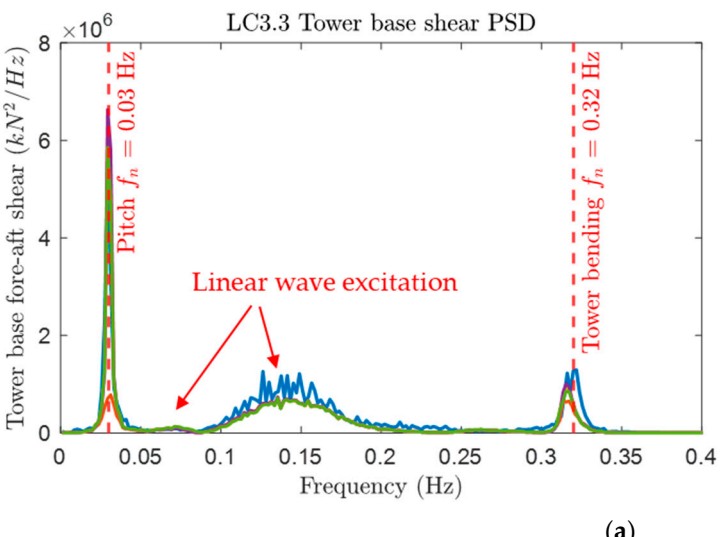
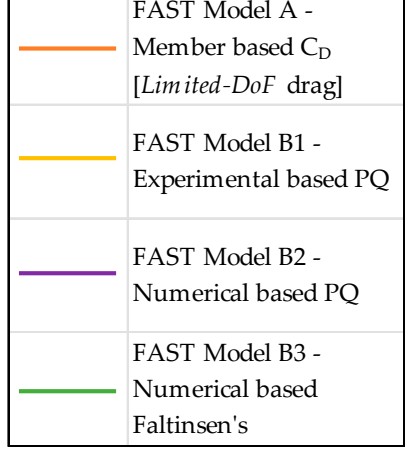

(**a**)

**Figure 7.** *Cont.*

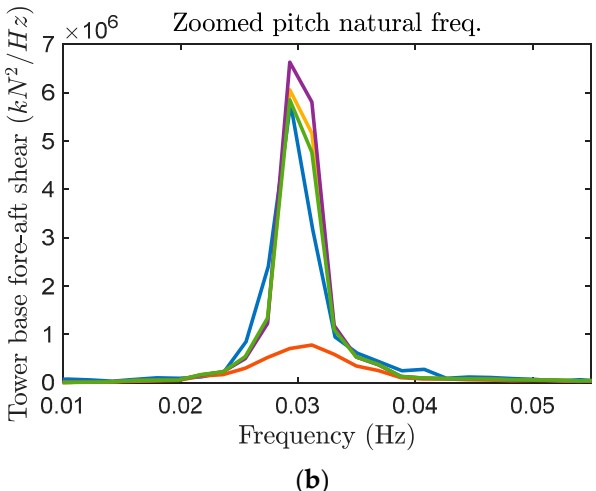

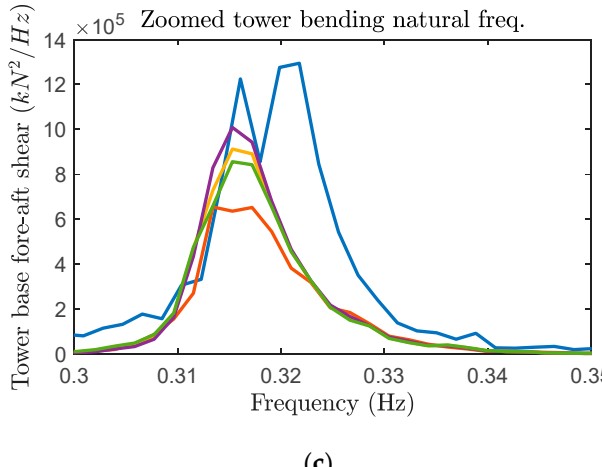

(**b**)

(**c**)

**Figure 7.** LC3.3 tower base fore-aft shear force power spectral density (PSD): (**a**) full spectrum view; (**b**) spectrum zoomed near pitch natural frequency; (**c**) spectrum zoomed near tower bending natural frequency.

At the frequency associated with pitch, Model A significantly underestimated the amplitude for low-frequency pitch when compared to the experiment, due to the use of member-based dimensionless drag coefficient. In contrast, the drag matrix models (B1-B3) only marginally overestimated the amplitude for low-frequency pitch. Model B2 produced the highest estimate, in which the drag matrix was tuned from Model A decay using the PQ method.

At the frequency associated with tower bending, all the models underestimated the response. Model B2 has the highest amplitude, followed by models B1, B3, and A.

### 4.2.3. PSD of Up-Wave Mooring Line Tension

For the LC3.3 irregular wave loading, Figure 8a plots the up-wave mooring line tension PSD. Similar to tower base fore-aft shear, sub-plots in Figure 8b,c zoom in on the two main regions of concern, namely near the surge and linear wave excitation frequencies.

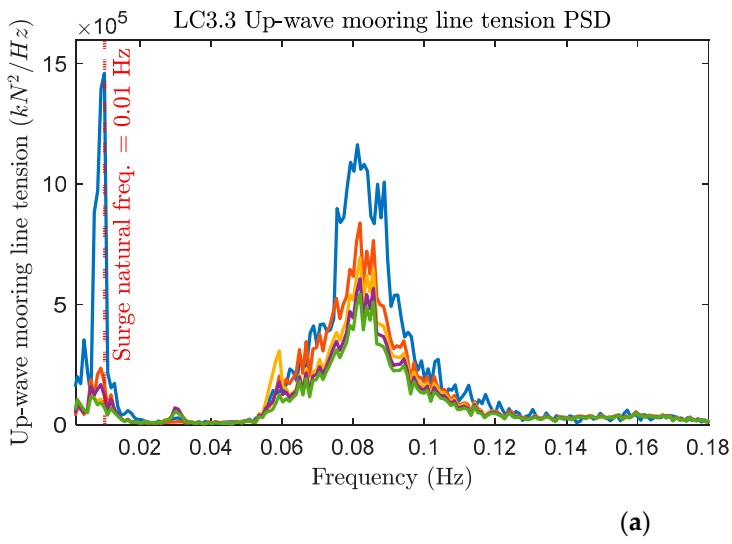

(**a**)

**Figure 8.** *Cont.*

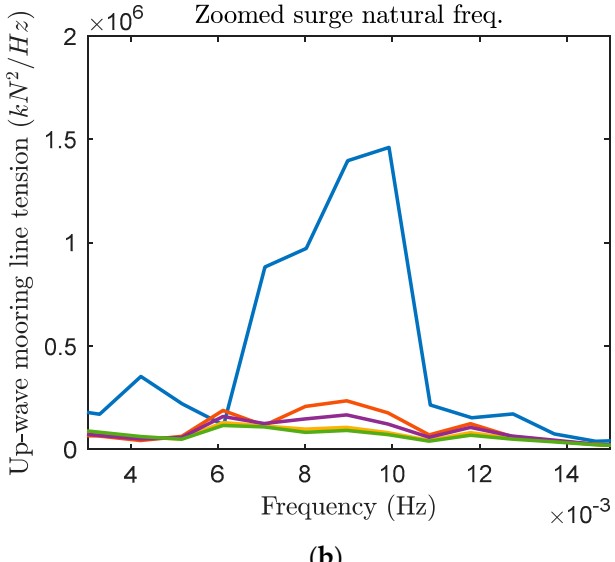

(**b**)

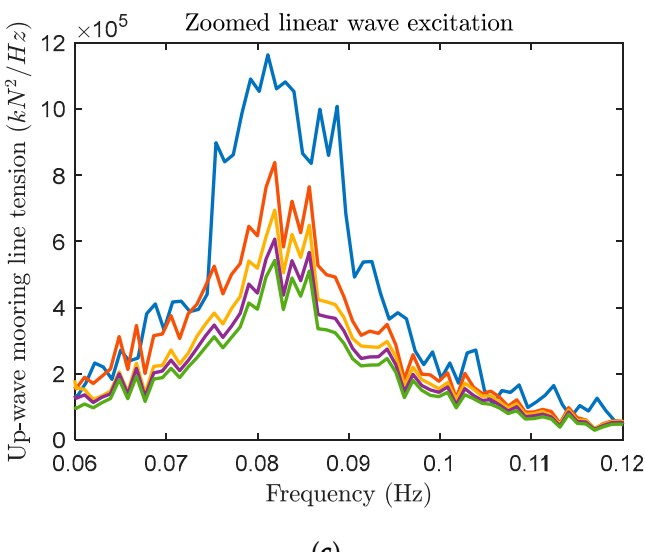

(**c**)

**Figure 8.** LC3.3 up-wave mooring line tension power spectral density (PSD): (**a**) full spectrum view; (**b**) spectrum zoomed near surge natural frequency; (**c**) spectrum zoomed near linear wave excitation.

All the numerical models clearly underestimated the amplitude at the frequencies associated with surge and linear wave excitation. Additionally, response was close together for all the numerical models. Model A had the largest amplitude among the numerical models, followed by models B2, B1, and B3. Trends observed here were similar to those observed for the surge-induced response, because the platform surge motion is a primary driver of up-wave mooring line tension.

A similar trend can be observed for the zoomed-in region for the linear wave excitation. All the numerical models underestimated the response amplitude and were close together. Model A has the largest response amplitude among the numerical models, followed by Models B1, B2, and B3. This trend is similar to the error percent results, wherein, the model error percent followed—Model A < B1 < B2 < B3. Note that large response amplitude corresponds to lower error percent. It can be thus inferred that the time-domain-based percent error results and the frequency-domain-based up-wave mooring line tension PSD plots yielded the same results, giving consistency and confidence to the approach.

## 5. Discussion

From the results, when the pitch drag is more properly represented, the platform pitch motion improves, as observed in the power spectral density plots. Irrespective of the type of excitation force, the platform pitch motion results in bending moment on the tower base due to the presence of a heavy rotor nacelle assembly (RNA) at the tower top of the FOWT, with induced shear in the tower base as well. Thus, models improving rotational DoF drag in pitch lead to significantly improved estimates of the tower base fore-aft shear, as shown by the 95-percentile of the peak load and spectral response. In particular, using the drag matrix models (B1, B2, and B3) resulted in near 7% to 10% improvements in estimates of the 95-percentile peak loads when compared to Model A for the tower base fore-aft shear.

Of all the numerical models, Model B2 (PQ method based) resulted in the smallest percent error (14%) when compared to the experimental results in terms of the 95-percentile peak loads of tower base fore-aft shear. Model B2 also provided the best estimates of the heave motion, closely followed by Model A, because the heave drag in Model B2 was tuned to the Model A decay, which was based on the experimental data. Thus, two of the main drivers for underestimating the 95-percentile peak loads of tower base fore-aft shear arise from mischaracterizing the rotational DoF quadratic pitch drag and translational DoF heave drag of the platform. With improved characterization of the platform pitch and heave drag, estimates of the tower base fore-aft shear loads were best between the

models. Note, all Model B variants (B1-B3) still possessed similar levels of 95-percentile peak loads and heave motion spectral response, with Model A and B2 resulting in the best estimate of response amplitude at the heave natural frequency. Thus, it can also be inferred that underestimates of the 95-percentile peak loads depend more on mischaracterizing rotational DoF pitch drag than mischaracterizing translational heave drag.

No significant differences were observed between the different drag matrix models, irrespective of the source and method of tuning. Between the drag matrix Model B variants, the numerical-based tuning proposed in Model B2 was at par or at times improved compared to the experiment-based tuning in Model B1 for the 95-percentile peak loads of tower base fore-aft shear. Note, this improvement was small. When compared to the 95-percentile peak load of the experiment, Model B2 resulted $\approx 2\%$ reduction in percent error (or improved load estimate) than Model A. Thus, tuning of Model B2 using the PQ method based on the free decay of a *limited-DoF* drag Model A, best characterized both rotational DoF pitch and translational DoF heave drag, leading to the best estimates of tower base fore-aft shear.

All the numerical models underestimated the 95-percentile peak load and spectral response of the up-wave mooring line tension. However, estimates were close together, indicating other modeling assumptions besides the drag model may also be at play. With respect to the experimental results, Model A resulted in a percent error of 5.11% and the drag matrix model B variants had a percent error in the range of $\approx 5.9-7.5\%$. The similarity suggests that surge damping and surge-affected second-order wave excitation forces were similar between Model A and the Model B variants.

Underestimates of the up-wave mooring line tensile forces could be alleviated through tuning the surge (transverse) quadratic drag [15,58] and/or tuning the second-order wave excitation forces [59] or by improving the numerical representation of the mooring system. The latter would address mischaracterizing the quadratic transfer functions (QTFs) generated in the potential flow solver WAMIT, leading to mischaracterizing wave mean and slow drift forces. To improve estimates of the second-order wave excitation forces, the QTFs from the potential flow solver could be better tuned using experiments or CFD to improve estimates of second-order wave excitation forces, assuming the effects of higher-order forces are negligible. A tuned QTF potential flow model with viscous drag accounted for by the quadratic drag matrix (tuned based on free decay) would likely result in better estimates of tower base fore-aft shear and up-wave mooring line tension peak loads along with their spectral response. The formulation of a numerical model with a tuned QTF and drag matrix is also independent of geometry, and thus versatile. These effects on surge-affected up-wave mooring line tension need to be studied in further detail.

## 6. Conclusions

This study aimed to numerically understand the impact of *limited* degree of freedom (DoF) drag coefficients on a floating offshore wind turbine (FOWT) simulation. Currently, the adequate representation of the hydrodynamic quadratic damping conditions for FOWT remains uncertain, because of its dependency on the type of application of the drag coefficients (dimensionless or not). To address these uncertainties associated with the FOWT hydrodynamic modeling, the open-source mid-fidelity numerical simulation tool, Open-FAST, was used to investigate and compare different hydrodynamic quadratic drag models against the experimental data from the Offshore Code Comparison, Collaboration, Continued, with Correlation (OC5) project. Two types of quadratic drag formulation were considered: (1) member-based dimensionless drag coefficient (*limited-DoF*) model and (2) quadratic drag matrix model (in dimensional form). The former is referred to as the Model A configuration, and the latter are the Model B variants, namely B1, B2, and B3, which differ in the source and method of tuning the drag matrix. Model B1 represents more conventional tuning of the drag matrix based on the experimental OC5 free decay data using the PQ method. Models B2 and B3 implement a novel tuning of the drag matrix based on the Model A decay data using the PQ and Faltinsen's methods, respectively.

Model A, which employed *limited-DoF* drag coefficients, consistently underestimated the 95-percentile peak loads and spectral responses for both tower base fore-aft shear and up-wave mooring line tension. In Model A, the Morison drag term is applied using the distributed strip theory and drag is calculated normal to the member orientation (in one direction). However, drag does affect multiple DoFs because rotational drag effects are approximated from translational drag coefficients. Explicitly defining rotational DoF drag, particularly pitch, using the drag matrix models enabled significantly improved estimates of the 95-percentile and spectral response for tower base fore-aft shear.

When the pitch drag was more properly represented, the platform pitch motion improved, leading to significantly improved estimates of the tower base fore-aft shear.

In comparing the Model A and Model B variants, mischaracterizing the rotational DoF quadratic pitch drag and translational DoF heave drag of the platform resulted in underestimates of the 95-percentile peak loads of tower base fore-aft shear. These underestimates depended more on mischaracterizing rotational DoF pitch drag than mischaracterizing translational heave drag.

Herein, valuable insight into the quantification of the effects of better characterization of pitch drag on a FOWT numerical simulation is presented. Novelty lies in the use of numerical models (Model B2 and B3) with drag coefficients tuned based on the *limited-DoF* drag model (Model A) free decay in addition to the experiment-based drag-coefficient-tuned model (Model B1). The proposed novel method of tuning (Model B2 and B3) produced global load approximations that are on a par or even improved (tower base fore-aft shear) compared to the conventional experimental free decay-based tuning (Model B1). Tuning of Model B2 using the PQ method based on the free decay of a *limited-DoF* drag (Model A) led to the best estimates of tower base fore-aft shear, because it best represented both pitch and heave drag.

All the numerical models underestimated the 95-percentile peak load and spectral response of the up-wave mooring line tension. However, estimates were close together, indicating that other modeling assumptions besides the drag model may also be at play.

Although this study highlighted differences between damping models, there is a limit to the capabilities of mid-fidelity numerical analyses. As they neglect some aspects of physical response to maintain computational efficiency, mid-fidelity numerical models, such as OpenFAST, must be calibrated with experimental data or higher fidelity computational fluid dynamics (CFD) models to best represent physical behavior. Within the hybrid hydrodynamic numerical model (potential flow theory with quadratic drag), there is a mischaracterization of both the drag force application and/or second-order wave excitation forces by the potential flow solver WAMIT. Herein, the drag force mischaracterization was studied in terms of the impact of *limited-DoF* drag. It is hypothesized that the drag models used herein alongside a better representation of second-order wave excitation forces would also result in improved numerical estimates, because surge, heave, and pitch motions are complementary to each other. Thus, future studies could improve the models presented herein even further with additional improvements to the potential flow model.

**Author Contributions:** Conceptualization, A.S. and B.R.; methodology, A.S. and J.B.G.; formal analysis, A.S., writing—original draft, A.S.; writing—review and editing, A.S., B.R., B.G.S., P.L. and J.M.B.I.; validation, P.L. and J.B.G.; visualization, A.S.; supervision, B.R., B.G.S. and P.L.; funding acquisition, J.B.G. All authors have read and agreed to the published version of the manuscript.

**Funding:** This work was partially supported by the Research Group of the Basque Government (IT1514-22) and the U.S. Department of Energy (DOE) under award number DE-EE0008960, titled "Coupled Aerodynamic and Hydrodynamic Hybrid Simulation of Floating Offshore Wind Turbines".

**Institutional Review Board Statement:** Not applicable.

**Informed Consent Statement:** Not applicable.

**Data Availability Statement:** Some or all data, models, or code that support the findings of this study are available from the corresponding author upon request. The findings, opinions, recommendations,

and conclusions in this paper are those of the authors alone and do not necessarily reflect the views of others, including the sponsors.

**Conflicts of Interest:** The authors declare no conflict of interest.

## Abbreviations

The following abbreviations are used throughout this manuscript:

| | |
|---|---|
| $A$ | Projected area |
| $B$ | Drag coefficients |
| $C$ | Additional linear stiffness |
| CFD | Computational Fluid Dynamics |
| CM | Center of Mass |
| CoV | Coefficient of variation |
| $C_D$ | Drag coefficient |
| $D$ | Cylinder diameter |
| DoF | Degree of Freedom |
| $F_{Add}$ | Generic Force Addition term |
| $F_{Drag}$ | Drag Force term |
| $F_0$ | Preload |
| FFT | Fast Fourier Transform |
| FOWT | Floating Offshore Wind Turbine |
| GWEC | Global Wind Energy Council |
| $H_{m0}$ | Significant Wave Height |
| IEA | International Energy Agency |
| L | Mooring line segments |
| $M$ | Mass matrix |
| MARIN | Maritime Research Institute Netherlands |
| $N$ | Total Signal Length |
| NREL | National Renewable Energy Laboratory |
| OC3 | Offshore Code Comparison Collaborative progr. |
| OC4 | Offshore Code Comparison Collaboration, Continued progr. |
| OC5 | Offshore Code Comparison Collaboration, Continued, with Correlation progr. |
| $p$ | Linear constituent coefficient |
| PQ | Linear (P) and Quadratic (Q) |
| PSD | Power Spectral Density |
| $q$ | Quadratic constituent coefficient |
| QTF | Quadratic Transfer Functions |
| $\Re$ | Real part of the complex number |
| $Rf$ | Autocorrelation function |
| RNA | Rotor Nacelle Assembly |
| $S_B$ | Body surface |
| SWL | Still Water Level |
| $T$ | Period |
| $t$ | Time |
| TCP | Technology Collaboration Program |
| TLP | Tension Leg Platform |
| $v$ | Fluid velocity |
| WAMIT | Wave Analysis at Massachusetts Institute of Technology |
| *Greek Symbols* | |
| $k$ | Hydrostatic stiffness coefficient |
| $\omega$ | Wave frequency |
| $\Phi$ | Generalized floating body motion in each DoF |
| $\phi$ | Crest amplitude value |
| % | Percent error |

**Appendix A. OpenFAST Hydrodynamic Model Characteristics**

The mid-fidelity FOWT numerical simulation tool, OpenFAST, was used to study the impact of *limited-DoF* drag coefficients. The characteristics of the hydrodynamic model used in the HydroDyn module of OpenFAST are summarized below.

- The hydrodynamic model uses the potential flow model (WAMIT solver) augmented with quadratic Morison drag terms or a $6 \times 6$ quadratic drag matrix, which allows for the inclusion of the radiation-diffraction solution and viscous drag (quadratic).
- In case of second-order wave kinematics, full sum and difference second-order WAMIT QTF is used. Wave stretching is not included. It is essential to include full difference QTF for moored FOWT analysis [28,39,44]. Even though the hydrodynamic model using full difference QTF underestimates the loads, those models are the closest to the experiment [28].
- Additional preloads ($F_0$ in N) in surge and sway were included using the generic force addition term $F_{Add}$, as shown below, to account for the influence of measurement cable bundle [42,44].

$$F_{Add} = F_0 = \begin{bmatrix} 1.40 \times 10^5 \\ 1.00 \times 10^4 \\ 0 \\ 0 \\ 0 \\ 0 \end{bmatrix} \tag{A1}$$

- Additional linear stiffness ($C$ in N/m) in surge and sway were included to account for the influence of measurement cable bundle using the generic force addition term $F_{Add}$ [42,44].

$$F_{Add} = [C]\Phi = \begin{bmatrix} 5000 & 0 & 0 & 0 & 0 & 0 \\ 0 & 7000 & 0 & 0 & 0 & 0 \\ 0 & 0 & 0 & 0 & 0 & 0 \\ 0 & 0 & 0 & 0 & 0 & 0 \\ 0 & 0 & 0 & 0 & 0 & 0 \\ 0 & 0 & 0 & 0 & 0 & 0 \end{bmatrix} \Phi \tag{A2}$$

- All the numerical models (Model A, B1-B3) have the inclusion of the same linear drag matrix to account for external damping from the experimental apparatus [42].

$$F_{Add} = [B_{linear}]\dot{\Phi} = \begin{bmatrix} 3.67 \times 10^4 & 0 & 0 & 0 & 0 & 0 \\ 0 & 1.60 \times 10^4 & 0 & 0 & 0 & 0 \\ 0 & 0 & 1.18 \times 10^4 & 0 & 0 & 0 \\ 0 & 0 & 0 & 7.68 \times 10^7 & 0 & 0 \\ 0 & 0 & 0 & 0 & 7.14 \times 10^7 & 0 \\ 0 & 0 & 0 & 0 & 0 & 6.34 \times 10^7 \end{bmatrix} \dot{\Phi} \tag{A3}$$

## Appendix B. Time Domain History Plots for Tower Base Fore-Aft Shear and Up-Wave Mooring Line Tension

LC3.3 Tower base fore-aft shear time domain history

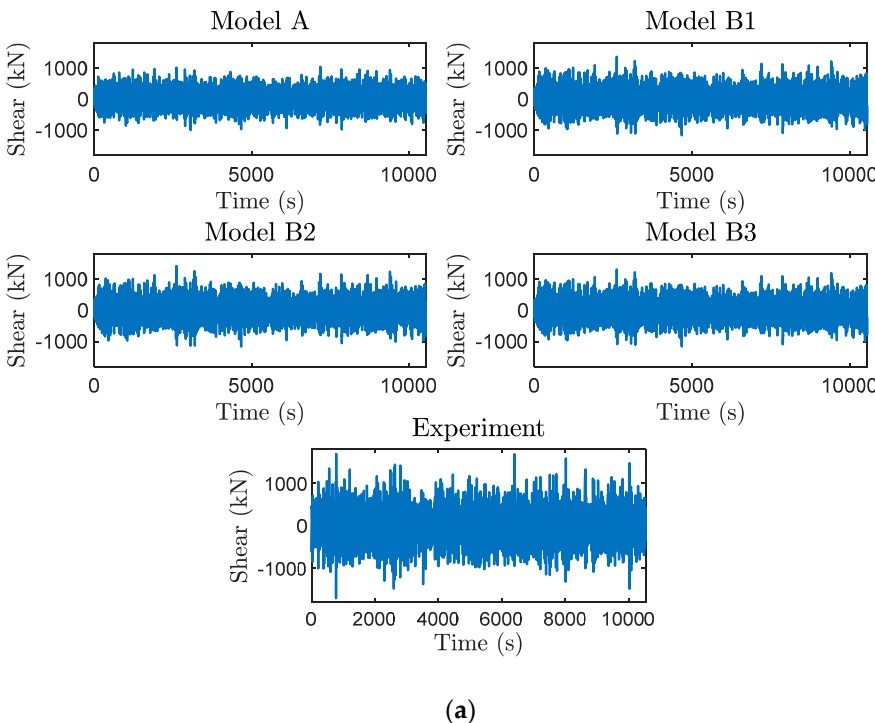

**(a)**

LC3.3 Up-wave mooring line tension time domain history

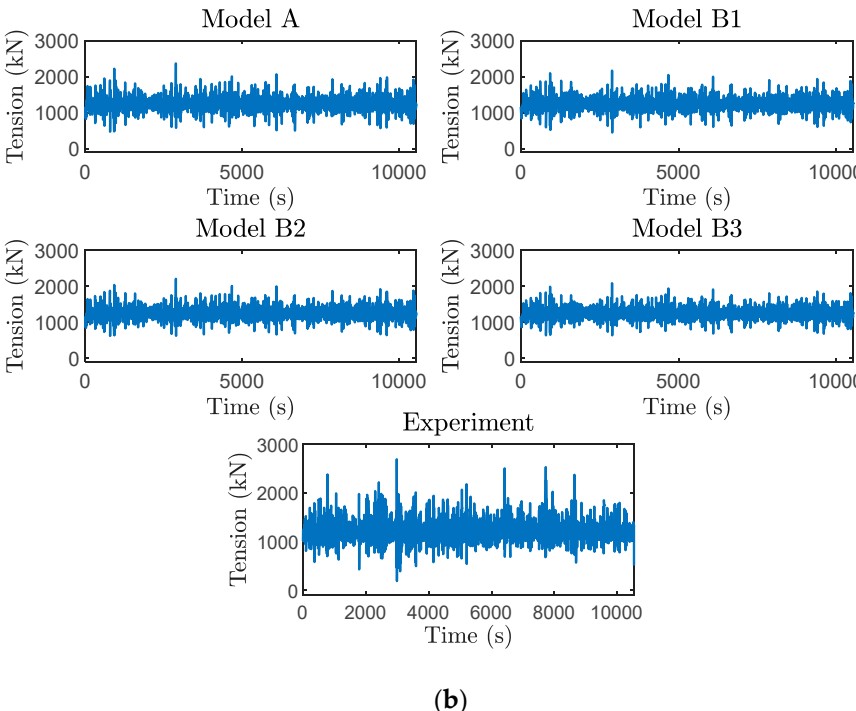

**(b)**

**Figure A1.** LC3.3 tower base fore-aft shear and up-wave mooring line tension time domain history: (**a**) tower base fore-aft shear time domain history; (**b**) up-wave mooring line tension time domain history.

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
