# Peer review of "Impact of Limited Degree of Freedom Drag Coefficients on a Floating Offshore Wind Turbine Simulation"

_jmse, doi:10.3390/jmse11010139_

Round 1

Reviewer 1 Report

Comments and Suggestions for Authors

Author Response

Revision and responses detailed in the attached document.

Reviewer 2 Report

Comments and Suggestions for Authors

Author Response

(The authors gave the same response as above.)

Reviewer 3 Report

Comments and Suggestions for Authors

The paper studies the effect of modelling damping in the skill of the precidctions of the motions of the Platform. The work is well described and the results obtained are discussed and conclusions stated. In general the paper is acceptable as is although minor revisons could improve it.

It mentions the platform used in the code comparison study, so it would make sense to refer to the results of that study and the uncertainty involved as discussed in:

doi10.1016/j.egypro.2017.10.333

https//doi.org/10.1016/j.oceaneng.2017.11.001

In line 141 please check automation in numbering of Table 1, so that it does not give error

In line 558 correct for the mention of Figure 7

Author Response

(The authors gave the same response as above.)

Round 2

Reviewer 2 Report

Comments and Suggestions for Authors

Accept in current form